# Mycovirus-Containing *Aspergillus flavus* Alters Transcription Factors in Normal and Acute Lymphoblastic Leukemia Cells

**DOI:** 10.3390/ijms251910361

**Published:** 2024-09-26

**Authors:** Cameron K. Tebbi, Jiyu Yan, Eva Sahakian, Melanie Mediavilla-Varela, Javier Pinilla-Ibarz, Saumil Patel, George E. Rottinghaus, Rachel Y. Liu, Clare Dennison

**Affiliations:** 1Children’s Cancer Research Group Laboratory, Tampa, FL 33613, USA; jiyuyanshine@gmail.com (J.Y.); rachelliu1957@gmail.com (R.Y.L.); 2Moffitt Cancer Center, Tampa, FL 33612, USA; eva.sahakian@moffitt.org (E.S.); melanie.mediavilla-varela@moffitt.org (M.M.-V.); javier.pinilla@moffitt.org (J.P.-I.); 3Tampa General Hospital, Tampa, FL 33606, USA; spatel@tgh.org; 4Electron Microscopy Department, University of Missouri, Columbia, MO 65211, USA; rottinghausg@missouri.edu; 5Diagnostic Laboratories, College of Veterinary Medicine, University of South Florida, Tampa, FL 33620, USA; clareldennison@gmail.com

**Keywords:** leukemia, acute lymphoblastic leukemia, cause of leukemia, etiology of leukemia, mycovirus, fungi, virus, *Aspergillus flavus*, genetics, transcription factors, environment, PAX5, IKAROS, NF-κB

## Abstract

Transcription factors control genes to maintain normal hemopoiesis, and dysregulation of some factors can lead to acute lymphoblastic leukemia (ALL). Mycoviruses are known to alter the genetics of their fungal host. The present study evaluates the effects of the products of a mycovirus-containing *Aspergillus flavus* (MCAF), isolated from the home of a patient with ALL, on certain transcription factors of normal and ALL cell lines. Our published studies have shown that ALL patients have antibodies to MCAF, and that exposure of the mononuclear leukocytes of patients in complete remission to its products, unlike controls, results in the re-development of genetic and cell surface phenotypes characteristic of ALL. For the present study, normal, pre-B, and B-cell leukemia cell lines were exposed to the culture of MCAF. Pre- and post-exposure levels of PAX5, Ikaros, and NF-κB were assessed. Exposure to MCAF resulted in apoptosis, cell cycle changes, and complete downregulation of all transcription factors in normal cell lines. In acute leukemia cell lines, cellular apoptosis and alterations in the cell cycle were also noted; however, while there was downregulation of all tested transcription factors, residual levels were retained. The noted alterations in the transcription factors caused by MCAF are novel findings. The possible role of MCAF in leukemogenesis needs to be further investigated. Mycovirus-containing *Aspergillus flavus* was initially isolated from a leukemia patient’s home. Our prior published studies have illuminated intriguing associations of this organism with leukemia. Unlike controls, patients diagnosed with acute lymphoblastic leukemia (ALL) harbor antibodies to this organism. Furthermore, the exposure of mononuclear cells from patients with ALL in complete remission to the products of this organism reproduced genetic and cell phenotypes characteristic of ALL. These findings underscore the potential role of environmental factors in leukemogenesis and hint at novel avenues for therapeutic intervention and preventive strategies.

## 1. Introduction

Leukemia ranks as the fifteenth most diagnosed human cancer and the eleventh cause of death due to a malignant disorder and affects both sexes and all age groups [1,2]. Acute lymphoblastic leukemia (ALL), with an incidence of 1.7 per 100,000 population per year, is seen at any age but is most commonly diagnosed in children, constituting 25–30% of all pediatric malignant disorders [1]. ALL originates from the transformation of B-cell progenitors. Many pre-B and B-cell acute leukemias have specific genetic changes and altered regulators of normal B-cell development. The exact mechanism by which these factors drive the leukemic transformation is unclear. Alteration in the normal differentiation process may be critical in the development of this disease [3,4,5,6].

Transcription factors have a vital role in normal lymphoid cell development. The actions of these factors include the integration of external signals to gene expression, programs that reconstruct cellular physiology at a basic level, and an array of post-transcriptional modifications (PTMs). The formation of lineage-restricted progenitors is a part of the function of several transcription factors that activate B-cell genes, including the restriction of alternative cell fates. Some factors reported to be consequential in promoting B cell differentiation include *PAX5*, *Ebf1*, and *Ikaros*.

PAX5 transcription factor, also known as B-cell specific activator protein (BSAP), is a master regulator of the normal development, differentiation, maturation, and maintenance of B-cells. Deletion or inactivating mutations of PAX5 results in chromosomal rearrangements, translocations, and cell arrest. The mutation of PAX5 is one of the consistent genetic alterations found in B-cell acute lymphoblastic leukemia (B-ALL). This transcription factor, and others, form a network that promotes B cell differentiation [4,7,8,9,10]. Experimentally and clinically, transcription factors have been found to play an essential role in ALL, where several of these factors are often downregulated [7,11]. Based on animal studies, the alterations and graded reduction of lineage are some of the factors that can induce leukemia [12,13]. In pre-B cell ALL, genes encoding some transcription factors are usually altered or deleted, indicating their role in this disease. It has been postulated that haploinsufficiency for *PAX5* or *Ebf1* synergizes with *STAT5* activation and can initiate the process of ALL development [14,15]. Furthermore, several common genetic variants associated with increased risk for ALL have been recognized [16,17,18]. In children with ALL, it is estimated that at least 4% are likely to have functional germline mutations [19,20]. In familial ALL, predisposing germline mutations in the hematopoietic regulator genes *PAX5, SH2B3, ETV6*, and *Ikzf1* have been reported [21,22,23,24,25,26,27]. Alteration in the *Ikaros* transcription factors also occurs in T-cell acute lymphoblastic leukemia [28]. It is proposed that the etiology of ALL involves a combination of genetic predisposition followed by a provoking event such as an infection [1,29,30,31,32,33,34]. We have previously shown that using the ELISA technique, patients with ALL in complete remission and long-term survivors have antibodies to the supernatant of the culture of a certain mycovirus-containing *Aspergillus flavus* (SMCAF) [35]. Multiple sets of controls, including normal individuals, patients with a variety of solid tumors, and those with sickle cell disease, lack such antibodies [35]. Interestingly, the organism used was isolated from the home of a patient with ALL. The isolated organism does not produce any aflatoxin, possibly due to its infestation with a mycovirus. Mycoviruses are shown to suppress the production of aflatoxin in their fungal host, which is attributed to the virus/virus and virus/host interactions [36,37,38,39]. Our previous studies have also revealed that exposure of the mononuclear cells from ALL patients in complete remission to the SMCAF results in the re-development of the cell surface phenotypes and genetic markers characteristic of ALL, while the controls remain unchanged [40].

Fungal agents in general, and the Aspergillus species in particular, are ubiquitous microorganisms, spread worldwide, and can be present indoors or outdoors. It has been estimated that between 30 to 80% of fungal species are infected with mycoviruses [41]. These are found to have the potential to cause significant changes in the genetics of their fungal host. Some mycoviruses have been shown to induce the transcriptional rewiring of their residing fungi [42]. The level expression of specific host genes differs in mycovirus-free and infected fungi. Alterations in transcription factors in mycovirus-infected fungi have been reported [43,44]. Experimentally, the transcriptome sequencing (RNA-seq) of mycovirus-infected *Malassezia sympodialis* has also been shown to result in an upregulation of several ribosomal components as compared to the virus-cured control, indicating that the mycovirus can modify the transcriptional and translational aspects of the host [42,45,46,47]. Only limited data regarding the effects of mycoviruses-infected fungi on human health are available.

The present study was designed to evaluate if the supernatant of a mycovirus-containing *Aspergillus flavus* (SMCAF) has any effects on the cellular transcription factors of pre-B and B-cell ALL compared to other leukemias and ‘normal’ control cell lines. The investigations also examine the impact of the SMCAF on the survival rate and cell cycle of established pre-B and B-cell ALL cell lines, compared to the ‘normal’ B-lymphocyte, T-cell leukemia, and chronic myelogenous leukemia cell lines.

## 2. Results

### 2.1. Chemical Analysis

Repeated evaluation of the supernatant of the culture of mycovirus-containing *Aspergillus flavus* revealed that this organism does not produce any aflatoxin.

### 2.2. Electron Microscopy

Transmission electron microscopy examination of the culture of *Aspergillus flavus* demonstrated the existence of the virus-like particles within the body of the organism and the culture supernatant (Figure 1). The sizes of the particles observed ranged between 30–50 nm and were in single or aggregate form, with or without patent dense cores. Particles ranging from 20–25 nm and 60–80 nm containing dense cores were seen in the hyphae.

### 2.3. Transcription Factors

Normal cell line: In the RPMI 1788 cell line, which was used as a ‘normal’ B-cell control cell line, the addition of SMCAF resulted in a dose-dependent downregulation of all transcription factors tested, i.e., *PAX5*, *Ikaros* 55 kD and 75 kD and *NF-κB p65*. No detectable levels remained with the 0.4 mg/mL (*p <* 0.01) (Figure 2, Figure 3, Figure 4 and Figure 5).

Pre-B and B-cell lines: A dose-dependent downregulation of *PAX5* in NALM6 clone G5 (CRL-3273) and GM20390 cell lines was noted (Figure 2). The pattern in the downregulation of *PAX5* was similar for the NALM6 and GM20390, pre-B and B-cell leukemia cell lines (Figure 2). However, unlike RPMI-1788, where doses of greater than 0.2 mg/mL of SMCAF resulted in the complete abolition of this transcription factor, in GM20390 and NALM6 cell lines, with the dose of 0.2 mg/mL, a statistically insignificant downregulation of PAX5 was noted (*p =* 0.23 and *p =* 0.65, respectively) (Figure 2). Even with doses of 0.4 mg/mL, while a statistically significant downregulation occurred (*p =* 0.006 and *p <* 0.0001, respectively), a residual PAX5 transcription factor could still be detected (Figure 2).

An evaluation of the effects of SMCAF on the *Ikaros* 55 kD transcription factor in NALM6 and GM20390 revealed that the downregulation was gradual and incomplete (Figure 3). This contrasts with RPMI-1788, a ‘normal’ cell line, where statistically significant downregulation of *Ikaros* 55 kD and 75 kD was noted by adding 0.1 mg/mL (*p =* 0.034 and *p =* 0.009, respectively). In the NALM6 cell line, with the doses of 0.1 to 0.3 mg/mL of SMCAF, downregulation of *Ikaros* 55 kD was not statistically significant (*p =* 0.9165, 0.8946 and 0.2725, respectively); however, with a dose of 0.4 mg /mL, the changes became substantial, but incomplete (*p =* 0.0411). In the GM20390 cell line, with doses of 0.1–0.4 mg/mL of the SMCAF, a gradual but incomplete downregulation of *Ikaros* 55 kD was noted (*p =* 0.0436, *p =* 0.0148, *p =* 0.0103, and *p =* 0.0091, respectively) (Figure 3). A similar finding was observed on the effects of SMCAF on *Ikaros* 75 kD on pre-B and B-cell lines, albeit with a more gradual downregulation of this transcription factor. In NALM6 and GM20390, statistically significant downregulation of *Ikaros* 75 kD was found with the addition of 0.3 and 0.4 mg/mL of SMCAF (*p =* 0.5787 and 0.0447 for NALM6; *p =* 0.0184 and 0.0238 for GM20390, respectively) (Figure 4).

Exposure of RPMI-1788 to SMCAF resulted in the downregulation of transcription factor *NF*-*κB p65* to the extent that no levels were detectable with SMCAF doses greater than 0.2 mg/mL (*p =* 0.0204). The effects were less significant in the pre-B cell NALM6 cell line, where only with SMCAF doses of 0.3 and 0.4 mg/mL did levels of this transcription factor significantly decline (*p =* 0.0014 and *p <* 0.0001, respectively). No significant changes of *NF*-*κB* found in GM20390 cells with any doses of SMCAF ranging from 0.1 to 0.4 mg/mL were noted (*p =* 0.5193, *p =* 0.9831, *p =* 0.5187 and *p =* 0.0625, respectively) (Figure 5).

T and CML Cell lines: The control cell lines Jurkat and K562 do not have the expression of the *PAX5* transcription factor. In the Jurkat cell line, upon exposure to the SMCAF, transcription factor *Ikaros* 55 kD downregulated; however, with doses of 0.1 to 0.2 mg/mL, this was not significant (*p =* 0.9016 and 0.3031, respectively). A considerable downregulation was noted with doses of 0.3 and 0.4 mg/mL (*p =* 0.0453 and *p =* 0.0214, respectively) (Figure 3). The transcription factor *Ikaros* 55 kD in K-562 cells showed no statistically significant changes with any doses of the SMCAF (Figure 3). In the Jurkat line, the addition of SMCAF resulted in a very gradual downregulation of the transcription factor *Ikaros* 75 kD, and only with doses of 0.3 and 0.4 mg/mL did the downregulation become significant (*p =* 0.0037 and *p =* 0.0003, respectively) (Figure 4). Similarly, the *Ikaros* 75 kD level in the K562 cell line was gradual, and it was only significantly downregulated with SMCAF doses of 0.3 and 0.4 mg/mL (*p =* 0.0425 and *p =* 0.0313) (Figure 4). The downregulation of *NF*-*κB p65* in the Jurkat cell line was only significant with SMCAF doses of 0.3 and 0.4 mg/mL (*p =* 0.0051 and *p =* 0.0003, respectively). In the K-562 cell line, with all SMCAF doses used, the *NF-κB p65* level was not significantly altered (*p >* 0.29) (Figure 5).

Culture media used as a control for SMCAF had no significant effects on the levels of any transcription factors in any of the cell lines tested (Figure 2, Figure 3, Figure 4 and Figure 5).

### 2.4. Apoptosis and Cell Cycle

Upon exposure to SMCAF, analyses revealed notable changes in the leukemic cell lines. The results of the apoptosis studies, conducted with annexin V/PI staining, showed that there was a significant increase in the early apoptosis in the GM20390, NALM6, and K562 cell lines after 72 h of treatment with the SMCAF (Figure 6A,B,D). In addition, a significant increase in late apoptosis was observed in the cell line GM16726 after the treatment (Figure 6C). Cell cycle analysis was performed after 72 h of supernatant in the culture of the mycovirus-containing *Aspergillus flavus* treatment. An increase in the G1 phase was observed in the GM20390 cell line (33.9% to 41.6%) (Figure 6E). In addition, an increase in the S phase was noted in the NALM6 (47.9% to 55.7%) cell line (Figure 6F). No changes were observed in the control cell lines. Corroboration of apoptotic cell death was observed while analyzing the cell cycle, in the increase in cells in G0 after treatment with the supernatant of the culture of mycovirus-containing *Aspergillus flavus* in NALM6 Jurkat cells. (Representative data from 2 independent experiments). Of course, the apoptosis and cell cycle studies were preliminary studies and not subject to statistical evaluations.

## 3. Discussion

The novel finding that products of a mycovirus-containing *Aspergillus flavus* can alter transcription factors in normal and leukemia cell lines is significant. The normal development, differentiation, and maintenance of the hematopoietic system require the activity of several signaling pathways, epigenetics, and transcription factors. These factors recognize and bind to specific DNA sequences and can control chromatin and transcription, resulting in a system that guides various genomic expressions. Transcription factors are essential in regulating cellular growth and developing hemopoietic cells. The chromosomal translocations and aberrant expression of transcription factors can potentially be associated with leukemogenesis. Transcription factors are shown to have a role in the cancer cell cycle, progression, metastasis, and resistance to treatment [48]. For example, the PAX5 gene, located at chromosome 9q13, is a member of the paired box (PAX) transcription factors that regulate early development and differentiation of B-cell lineage in the hemopoietic system. This transcription factor activates B cell-specific genes and represses those specific for other hematopoietic lineages. PAX5 regulates its target genes via recruiting chromatin-modifying proteins in the committed B cells. The PAX5 gene encodes the B-cell lineage-specific protein (BSAP), expressed during the early stages of B-cell development. Alteration in PAX5 has been implicated in developing human B-cell malignancies, including precursor B-cell acute lymphoblastic leukemia.

Our findings, for the first time, indicate that exposure to mycovirus-containing *Aspergillus flavus* can alter the transcription factors. In the described experiments, the exposure altered most transcription factors and resulted in cellular apoptosis and cell cycle alteration. These novel findings may indicate that mycovirus-containing fungi may have the potential to modulate the rate of gene transcription, altering cellular division, proliferation, differentiation, metabolism, function, and apoptosis. These findings may have therapeutic and etiological significance. In our experiments, changes in the transcription factors were not uniform, and the effects significantly differed between the ‘normal’ and leukemia cell lines. Significant differences in acute versus chronic leukemias were noted within the leukemia cell lines. In the *RPMI-1788*, which was used as a ‘normal’ control, exposure to lower doses of SMCAF resulted in the downregulation of all transcription factors tested to the point of non-detection (Figure 2, Figure 3, Figure 4 and Figure 5). In contrast, even with the highest dose utilized, some transcription factors were detectable in all leukemia cell lines. Interestingly, the effects of the supernatant of the mycovirus-containing *Aspergillus flavus* on the transcription factors, such as *NF-κB p65*, were significantly different in acute and chronic leukemia cell lines. In the chronic myelogenous leukemia cell line, K-562, used as a control, transcription factors were still detectable even with the highest doses of SMCAF. In this cell line, *NF*-*κB p65*, downregulation with all the doses used was not statistically significant.

Specifically, in the pre-B and B-cell lines, the downregulation of *PAX5* was only seen with much higher doses of SMCAF compared to *RPMI-1788*, a ‘normal’ control cell line. In the pre-B and B-cell leukemia cell lines, with doses of 0.1–0.2 mg/mL, statistically significant downregulation of *PAX5* was not noted; even with the highest dose of SMCAF, this transcription factor was still detectable. Located on chromosome 9 p13, *PAX5* is known to act as a transcriptional activator or repressor of the genes involved in the B-lineage development. PAX5 functions as a master regulator and a factor for IgH locus rearrangement. In addition, it inhibits the differentiation toward other lineages. The CD19 expression, which occurs in the later stages of B-cell development, is controlled by this transcription factor. Alterations and loss of the *PAX5* are suggested to contribute to leukemogenesis [14]. Loss-of-function mutations in *PAX5* transcription factor occur in B-progenitor acute lymphoblastic leukemia [49,50,51]. In childhood B-cell progenitor ALL, genome-wide analysis using oligo SNP arrays, *PAX5* was found to be the main target of somatic mutations, which was altered in 38.9% of the cases [8]. In one adult study of ALL, PAX5 was found to be mutated in 34% of the patients [49]. The mutation reduces levels of *PAX5* protein or the generation of hypomorphic alleles [8]. This is due to a partial, rather than complete, loss of function of this transcription factor. In some studies, a *PAX5* fusion product, P5-C20orf112, induces downregulation of pre-B cell receptor genes and causes differential proliferation patterns in B-cell lymphoblastic cell lines [52,53,54]. *PAX5* is found to be involved in several leukemia-associated rearrangements, resulting in the fusion genes encoding chimeric proteins that antagonize *PAX5* transcriptional activity. It is shown that individuals with loss-of-function variants and those with somatic deletion of the wild type of *PAX5* allele can develop ALL [55]. Therefore, the downregulation of this transcription factor upon exposure to SMCAF is a significant finding.

In a reported study of a murine model, transgenic RNAi was used to reversibly suppress endogenous *PAX5*, which cooperates with an activated signal transducer and activator of transcription 5 (STAT5) to induce B-ALL [14]. In this model, the restoration of endogenous *PAX5*, even temporarily, reversed the process, allowing for the surface expression of mature B cell markers and release of the pre-B stage differentiation block. This and similar studies have established a contributory relationship between the PAX5 and the development of pre-B ALL [14,52,56]. In this regard, the data presented in this manuscript, which for the first time show that exposure to SMCAF results in the downregulation of this transcription factor, is of importance.

In our studies, *Ikaros* transcription factors were tested in all our experimental cell lines, as these are essential regulators of lymphopoiesis and are known to be involved in ALL [57]. Upon exposure to the products of MCAF, the downregulation of *Ikaros* 55 kD/*Ikaros* 75 kD was seen to occur in all cell lines tested. In RPMI-1788, a ‘normal‘ cell line, with the addition of 0.1 mg/mL or a higher amount of SMCAF, the downregulation of *Ikaros* 55 kD and 75 kD were statistically significant. Higher doses resulted in the total elimination of these factors (Figure 3 and Figure 4). Downregulations in the leukemia cell lines were less intense and were, even with the largest dose of 0.4 mg/mL, incomplete. In GM20390 and NALM6 clone G5 (CRL-3273), the downregulation of *Ikaros* 55 kD occurred after exposure to doses greater than 0.1 and 0.4 mg/mL of SMCAF, respectively (Figure 3). Likewise, statistically significant downregulation of *Ikaros* 75 kD was found with 0.2 and 0.3 mg/mL of SMCAF, respectively (Figure 4). In these cell lines, *Ikaros* 55 kD, and more noticeably, *Ikaros* 75 kD, were still detectable even with 0.4 mg/mL, which was the highest dose used. The effect of SMCAF on the Jurkat, and K-562 cell lines was less intense than those of the pre-B and B-cell lines and did not result in the total elimination of the *Ikaros* 55 kD and 75 kD transcription factors, with a significant residual detectable with the highest dose utilized.

Our finding demonstrates for the first time that the product of mycovirus-containing *Aspergillus flavus* can alter and downregulate *Ikaros* 55 kDa and 75 kDa. *Ikaros*, a zinc finger transcription factor, is a significant hematopoiesis regulator and is frequently deleted or mutated in B-cell precursor acute lymphoblastic leukemia [58,59,60]. Somatic mutations or an alteration of *Ikzf1* is seen in 15–20% of childhood B-cell ALL; its deletion is seen in over 75% of BCR-ABL positive disease [8,57,60,61,62,63,64,65]. *Ikzf1* mutations occur in up to approximately 50% of adults with ALL [66,67]. The mechanisms involved in *Ikaros* regulation of the gene expression and cellular proliferation in T-ALL are unknown. It has been shown that the reintroduction of *Ikaros* into *Ikaros*-null T-ALL cells result in the termination of cellular proliferation and the induction of T-cell differentiation [68].

As transcription factors, *Ikaros* are critical regulators of lymphocyte ontogeny and differentiation [69,70,71,72,73,74]. During hematopoiesis, Ikaros functions as a transcriptional activator or repressor for B and T cell differentiation by recruitment of chromatin remodeling complexes [69]. The IKZF1 gene encodes the Ikaros protein, a hemopoiesis regulator. It is essential in developing all lymphoid lineages and functions as a tumor suppressor [70].

In our studies, upon exposure to SMCAF, downregulation of *NF*-*κB* was noted, on a dose-dependent basis, in all cell lines, except K562 (Figure 4). This downregulation was most notable in RPMI-1788, where smaller doses of SMCAF resulted in the total elimination of *NF*-*κB*. In the Pre-B and B-cell cell lines, while the levels of this transcription factor steadily declined with doses of 0.3 mg/mL and 0.4 mg/mL of SMCAF (*p =* 0.062 and *p =* 0.001, respectively), it was not completely eliminated with the highest doses utilized. Of interest is that, in the K562, a CML cell line, there was no statistically significant decline in the levels of *NF-κB p65*, even with the highest dose of SMCAF, i.e., 0.4 mg/mL used (Figure 4). Constitutive activation of *NF*-*κB* complexes has been reported in most (39/42) ALL patients without subtype restriction [9].

*NF*-*κB* and its associated regulatory factors are involved in cell proliferation and in the control of apoptosis in leukemias [75,76,77,78,79]. Given the role of the *NF*-*κB* pathway in leukemia, it is essential to investigate the inhibition of its activity by agents capable of blocking its pathway. NF-κB inhibitory molecules may be used, singularly, or in combination with chemotherapeutic agents, to treat hematological malignancies [80,81,82,83]. The finding of the downregulation of *NF*-*κB*, *PAX5*, and *Ikaros* 55 and 75 kD under the influence of SMCAF in certain acute leukemia cell lines is of significance. The fact that SMCAF can change various transcription factors in these cell lines is novel and is of value. This, along with our prior published findings revealing that patients with B-cell ALL, in complete remission and long-term survivors, unlike controls, uniformly have antibodies to SMCAF, and that the exposure of their mononuclear leukocytes to these products results in the re-development of characteristic genetics and cell surface phenotypes of this disease, is of interest and requires further investigation [35,40]. In light of the hypotheses indicating that the development of ALL is due to a combination of the genetic predisposition followed by a provoking event such as an infection [1,13,29,30,31,32,33,34], the role of a mycovirus-containing *Aspergillus flavus* in leukemogenesis needs to be further investigated.

Microorganisms have been implicated in the etiology of malignant disorders, mainly due to their various effects resulting in genetic or epigenetic changes [84,85,86,87]. It has been estimated that infectious microorganisms cause 18% of all malignant disorders [88]. This is more pronounced in developing countries, where 26% of cancers are attributed to infections compared to 8% in developed nations [89,90,91,92,93]. Some viruses are known to have the ability to alter the genetics of their host as a part of their cytopathogenesis. The incorporation of the viral genome into the host chromosome can be incidental or occur as part of the life cycle of these organisms. Viral genome integration can potentially lead to significant cellular consequences, including gene disruption, insertional mutagenesis, oncogenesis, and apoptosis [94,95,96]. As noted previously, mycoviruses can alter their fungal host’s phenotype, including the host’s pigmentation, morphology, sexual and asexual sporulation, production of aflatoxin, and growth. Some dsRNA mycovirus-containing fungal agents have been shown to alter the expression of genes involved in the ribosomal synthesis and programmed cell death of the fungal host. Whether these organisms can exert any changes in humans or animals infected with mycovirus-containing fungi has not been explored and requires investigation.

In the past, the carcinogenic effects of fungi have been generally attributed to their mycotoxin production, and in the case of *Aspergillus* species, to the aflatoxin [36,93,97]. *Aspergillus* infected with a mycovirus, as is the case in that investigated in the above studies, does not produce any aflatoxin. This is attributed to the mycovirus infection, the removal of which results in the reproduction of aflatoxin. Indeed, this phenomenon is used to take advantage of controlling aflatoxin in grain crops. Therefore, the effects of up- and downregulation of the transcription factors in our experiments cannot be attributed to aflatoxin. Rare reports of mycovirus-containing fungi affecting humans as a pathogen and their effects on the infected individuals are available. An example is the Malassezia species, which produces various skin diseases including dandruff, seborrheic dermatitis, and atopic dermatitis. In one study, this organism contained MrV40 mycovirus, which belongs to the Totiviridae family [98].

Prior attempts to alter transcription factor activities for therapeutic purposes have been made in several types of cancer by direct mechanisms such as amplification or deletion of genes, point mutations, chromosome translocations, and alteration of expression, or indirectly via non-coding DNA mutations, which can alter binding of the transcription factor. Attempts to change the mode or levels of transcription factors for therapeutic purposes also have been made. This has included blockage of transcription factor–cofactor protein–protein interactions, the prevention of transcription factor–DNA binding, and modulating the levels of transcription factors. The latter has been attempted by altering levels of ubiquitylation and ensuing proteasome degradation or by inhibiting regulators of transcription factor expression [99]. Other efforts to change the transcription factors have included targeting small-molecule-based heterobifunctional proteolysis targeting chimera (PROTACs), which modulates protein target levels by taking over the ubiquitin–proteasome system to bring about degradation of the target [100,101,102]. Several new approaches aimed at transcription factors have recently emerged. These include, as follows: PROTAC-mediated protein degradation; modulation of auto-inhibition; the use of cysteine reactive inhibitors targeting intrinsically disordered regions of transcription factors; and combinations of transcription factor inhibitors with kinase inhibitors to block the development of resistance.

In the light of prior findings showing that patients with ALL have antibodies to the SMCAF and that exposure to this product can cause the re-development of cell surface phenotypes in those in remission, but not in controls, the alteration of the transcription factors noted in this report is of significance. Further investigation of the possible relationship between mycovirus-containing filamentous fungi and leukemogenesis is warranted.

## 4. Materials and Methods

### 4.1. Mycovirus-Containing Aspergills flavus

This organism was isolated from the home of a patient with ALL and cultured in an underlayer of 1% solid agar with an overlayer of 3.5% Czapek-Dox broth (Difco; Becton Dickinson, Sparks, MD, USA) in a glass bottle, incubated at 28 °C and sub-cultured at approximately four-week intervals. To ensure the persistence of the mycovirus in the *Aspergillus flavus*, the cultures were periodically checked by transmission electron microscopy. For the described studies, a portion of the supernatant of mycovirus-containing *Aspergillus flavus* was collected after approximately four weeks of culture, filtered through a 0.45 µm filter (Thermo Scientific, Swedesboro, NJ, USA. Catalogue #169-0045), and concentrated via a centrifugal filter device with 3K of nominal molecular weight limit (NMWL) (Amicon Ultra-15 centrifugation device, EMD Millipore, EMD Millipore Corporation, Taunton, MA, USA) at 4000× *g* for 55 min. The concentrated SMCAF was quantitated by a BCA protein assay kit (Thermo Fisher Scientific, Pittsburgh, PA, USA) and had an approximate total protein concentration of 3–4 mg/mL.

Cell lines were initially obtained from the Coriell Institute for Medical Research (Camden, NJ, USA). The pre-B and B-cell ALL cell lines were GM20390 and NALM6 clone G5 (CRL-3273). For comparison, RPMI-1788, which is a ‘normal’ cell line, BCL2 Jurkat, an acute T-cell lymphoblastic leukemia cell line, and K562-S (CRL-3343), a chronic myelogenous leukemia (CML) cell line, were used. All cell lines were cultured in RPMI-1640 (Thermo Fisher Scientific, Waltham, MA, USA) with 10% fetal calf serum (Thermo Fisher Scientific, Waltham, MA, USA) and penicillin/streptomycin antibiotics and incubated at 37 °C with 5% CO_2_. Cells were harvested, counted, adjusted, and seeded at 2.4 × 10^4^/mL for culture. For the described studies, each cell line was cultured for three days with concentrated SMCAF using 0.1, 0.2, 0.3, and 0.4 mg SMCAF protein per milliliter. In a limited study, the effects of doses ranging from 0.01 to 0.05 of SMCAF were initially tested using the same condition to evaluate the impact of smaller doses. The culture media was used as the control. Upon harvest, cell count and viability were tested for each culture performed. For measurement of the cell viability rate, pre- and 72 h post-treatment of each cell line cultured with SMCAF were stained with methylene blue and counted using a hemocytometer. The percentage of survival was recorded.

### 4.2. Electron Microscopy

To evaluate the *Aspergillus flavus* for the presence of mycoviruses, the culture’s fungal growth and supernatant were analyzed for viral contents by electron microscopy. The culture-grown organism was fixed in glutaraldehyde and osmium tetroxide. Before transmission electron microscopy observation, this was placed into resin blocks suitable for ultramicrotomy sectioning at 70 nm, collected on copper grids, and contrast-enhanced with uranyl acetate.

### 4.3. Cell Cycle Studies

For the cell cycle and apoptotic studies, GM16726, which is an acute lymphoblastic leukemia cell line with lymphomatous features, was used. Cell cycle analysis studies were performed to characterize the effect of the SMCAF on the cells. Cells were washed and resuspended in 200 μL of PBS, followed by the dropwise addition of 2 mL of ice-cold 70% ethanol during vortex mixing. The cell suspension was incubated at −20 degrees centigrade for two hours. The cells then were washed with PBS and resuspended in 400 μL of a staining solution containing 0.6 mM of propidium iodide (PI) (Invitrogen P3566, Invitrogen, Carlsbad, CA, USA), 0.2 mg/mL of RNAse (ThermoFisher, R1253, ThermoFisher Scientific, Pittsburgh, PA, USA) and 0.1% *v*/*v* of Triton X100 (ThermoFisher, BP151-100, ThermoFisher Scientific, Pittsburgh, PA, USA). Cells were incubated at 37 degrees centigrade for 30 min before measurement of fluorescence using an LSR II flow cytometer (BD Bioscience, Franklin Lakes, NJ, USA). Data were analyzed using FlowJo 10.7 software (Tree Star Watson model, Tree Star, Inc., San Carlos, CA, USA).

### 4.4. Annexin V/PI Analysis

To examine the apoptotic cell death, cells were seeded at 2 × 10^5^/well in a T75 flask in RPMI 1640 media with 10% fetal calf serum and 1% Penicillin/Streptomycin and incubated at 37 °C with 5% CO_2_. The cells were treated with SMCAF with protein concentrations, as outlined above. Cultures were collected after 72 h of incubation. Cells were resuspended in 100 μL of Annexin V binding buffer (component no. 51-66121E, BD Biosciences, San Diego, CA, USA) with 5 μL of Annexin V-FITC (component no. 556419, BD Biosciences, San Diego, CA, USA) and incubated for 15 min at room temperature. At this point, 200 μL of Annexin V staining buffer was added, and fluorescence was measured using an LSR II flow cytometer (BD Biosciences, San Diego, CA, USA) and data were analyzed using FlowJo software (Tree Star, Inc., San Carlos, CA, USA). Annexin V positive, PI negative cells were identified as early apoptotic while Annexin V positive, PI positive cells were identified as late apoptotic.

### 4.5. Analysis for Detection of Aflatoxin

For the detection of aflatoxin in the supernatant of the culture of the mycovirus-containing *Aspergillus flavus*, a method commonly used for the quantitation of this toxin in cereal grains and complex feedstuffs was used. Samples were extracted with acetonitrile-water (7:3). The extracts were passed through EASI-EXTRACT aflatoxin immunoaffinity cleanup columns. The HPLC analyzed samples with Kobra cell post-column derivatization with fluorescence detection at 365 nm excitation and 440 nm emission. The detection limit was set at ten ppb total aflatoxins. It should be noted that periodically, the SMCAF was tested to ensure that it did not contain any aflatoxin.

### 4.6. Western Blot

The levels of transcription factors *PAX5*, *Ikaros* 55 kD and 75 kD, and *NF-κB p65* were measured, with and without exposure to SMCAF, using an immunoblotting technique. For Western blot, each cell line was harvested and centrifuged at 2000 RPM at 20 °C for five minutes and the pellets were washed twice with 4 mL of ice-cold Tris-buffered saline (TBS) (ThermoFisher Scientific, Pittsburgh, PA, USA). Cells were treated with a specified amount of SMCAF, as described above, or culture media, which was used as a control. Radio-immune precipitation assay (RIPA) lysis buffer (ThermoFisher Scientific, Pittsburgh, PA, USA) was used at a final 3 × 10^7^ cells/mL concentration. The RIPA consisted of 50 mM Tris–HCl (pH 7.4), 1.0% NP–leupeptin, and pepstatin (ThermoFisher Scientific, Pittsburgh, PA, USA). Each cell lysate was mixed on a shaker for 15 min at 4 °C, sonicated twice for 10 s at 50 kHz, shaken for 15 min on ice, and then centrifuged at 12,000× *g* at 4 °C for 20 min. The total protein was measured using a bicinchoninic acid (BCA) assay, aliquoted in 10 µg/µL in the loading buffer, and then denatured at boiling water for five minutes before being subjected to the Western blot analysis. The precast mini tris–glycine gel 4–20% (Bio-Rad, Hercules, CA, USA) was utilized to perform protein electrophoresis. A 0.22 µm nitrocellulose membrane and Efficient Western Transfer Buffer (Bioscience, St Louis, MO, USA) were used for protein transfer. 5% dry milk in 1x TBST wash buffer (tris-buffered saline with 0.05% Tween 20) was utilized for the membrane blocking. To detect transcription factors, *Ikaros*, *NF-κB p65*, and *PAX5*, appropriate primary monoclonal antibodies, and the secondary antibody HRP-linked anti-rabbit IgG were used (Cell Signaling Technology, Danvers, MA, USA). The enhanced chemiluminescence system (Viagene Biotech, Tampa, FL, USA) detected specific antibody binding and was read on the FlourChem E system (ProteinSimple, San Jose, CA, USA). The Western blot images were analyzed using FIJI/Imaging V1.52P-Win 64 software. Each study was repeated four times, and statistical analyses were performed based on these repeats. The chemiluminescent signal intensity for each transcription factor was measured and divided by the signal intensity for actin in the same blot lane to arrive at a transcription factor/actin ratio. These values were normalized to a NALM6 control sample on each blot to yield relative chemiluminescence values. These values were graphed using GraphPad Prism 8 software with representative blot images included in the figures.

### 4.7. Statistical Analysis

The GraphPad Prism two-tail unpaired t-test between control and SMCAF treatment groups was used for statistical analysis of the data. The statistical significance was set at *p*-value < 0.05. 

## Figures and Tables

**Figure 1 ijms-25-10361-f001:**
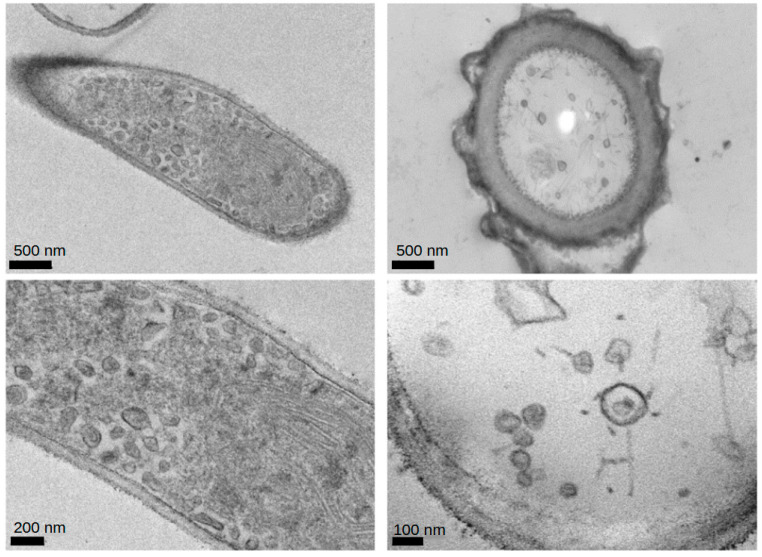
Viral particles measuring 20–80 nm in diameter found in the transmission electron microscopy evaluation of the culture of *Aspergills flavus* isolated from home of a patient with acute lymphoblastic leukemia. The sample was collected from the suspension culture of *Aspergillus flavus* and was glutaraldehyde-fixed. Transmission electron microscopy is a micrograph of a formvar-coated copper grid preparation that is contrast-enhanced with aqueous uranyl acetate. Scale bars are indicated at the bottom of each panel.

**Figure 2 ijms-25-10361-f002:**
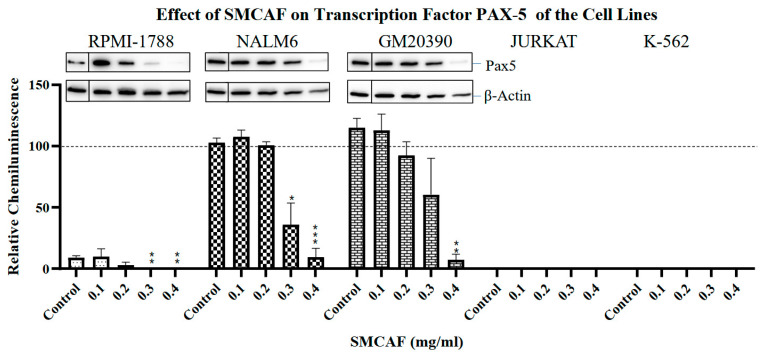
The effects of mycovirus-containing *Aspergillus flavus* on the ‘normal’ control cell line (RPMI-1788), pre-B (NALM6) and B-cell (GM20390) cell lines. Basal levels of the *PAX5* transcription factor, which are lower in the ‘normal’ cell line, completely receded at the highest dose of SMCAF. PAX5 basal levels are higher in pre-B ALL and B-cell ALL lines and are significantly downregulated by SMCAF, but not completely lost. Jurkat and K-562 cells did not express PAX5 under any conditions tested. Error bars represent the mean (SEM) standard error for three replicates. The dotted line represents the normalized value of 100 for a NALM6 control sample from each blot. Asterisks denote statistical significance relative to its own control (* *p* < 0.05; ** *p* < 0.01, *** *p* < 0.001). For each cell line, Western blot images include the control bands in a separate box from the treated bands; these images are from the same blot and exposure but reflect the cropping out of additional lines that were not used in this evaluation.

**Figure 3 ijms-25-10361-f003:**
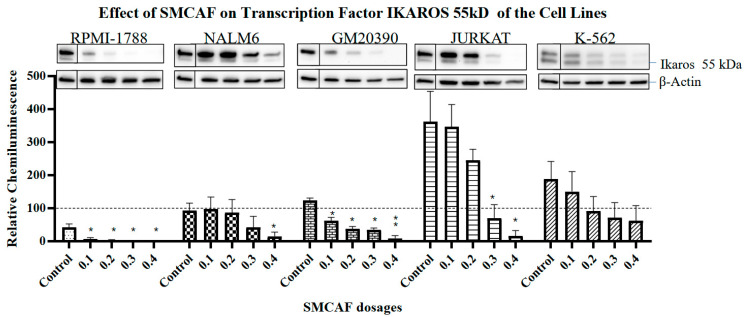
The effects of mycovirus-containing *Aspergillus flavus* on the ‘normal’ control B-cells (RPMI-1788), pre-B ALL (NALM6), B-cell ALL (GM20390), T-cell leukemia (Jurkat) and CML (K-562) cell lines. In the *Ikaros* 55 kDa transcription factor is downregulated in a dose-dependent manner and is completely eradicated at the highest dose of SMCAF. *Ikaros* 55 kDa basal levels are highest in Jurkat cells and downregulated in a dose-dependent manner in pre-B ALL, B-cell ALL, T-cells, and CML cells. Error bars represent standard error of the mean (SEM) for three replicates. The dotted line represents the normalized value of 100 for a NALM6 control sample from each blot. Asterisks denote statistical significance relative to its own control (* *p <* 0.05; ** *p <* 0.01). For each cell line, Western blot images include the control bands in a separate box from the treated bands; these images are from the same blot and exposure but reflect the cropping out of additional lines that were not used in this evaluation.

**Figure 4 ijms-25-10361-f004:**
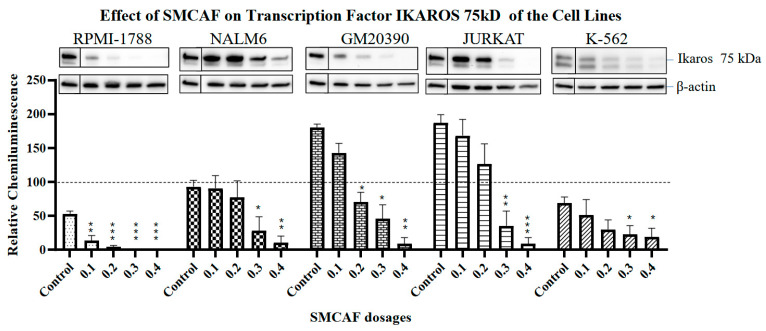
The effects of mycovirus-containing *Aspergillus flavus* on ‘normal’ control (RPMI-1788), pre-B (NALM6), B-cell (GM20390), T-cell leukemia (Jurkat) and CML (K-562) cell lines. Basal levels of the *Ikaros* 75 kDa transcription factor are highest in the T-cell line. *Ikaros* 75 kDa is downregulated in a dose-dependent manner in all five cell lines. Error bars represent the standard error of the mean (SEM) for three replicates. The dotted line represents the normalized value of 100 for a NALM6 control sample from each blot. Asterisks denote statistical significance relative to its own control (* *p <* 0.05; ** *p <* 0.01, *** *p <* 0.001). For each cell line, Western blot images include the control bands in a separate box from the treated bands; these images are from the same blot and exposure but reflect the cropping out of additional lines that were not used in this evaluation.

**Figure 5 ijms-25-10361-f005:**
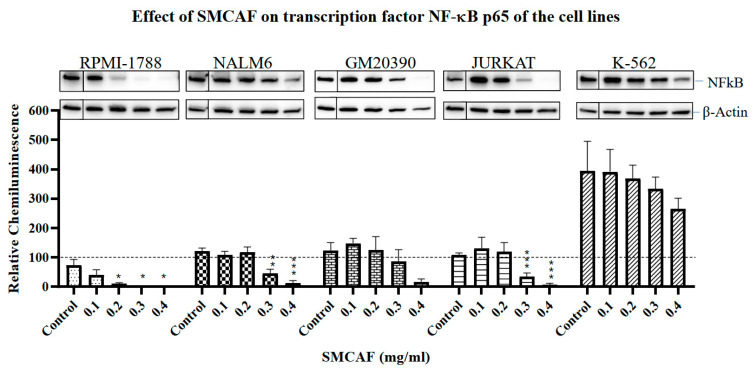
The effects of mycovirus-containing *Aspergillus flavus* on ‘normal’ control (RPMI-1788), pre-B (NALM6), B-cell (GM20390), T-cell leukemia (Jurkat) and CML (K-562) cell lines. While the downregulation of *NF-κB p65* transcription factor in the ‘normal’ cell line is complete with no residual remaining, this is significantly less and incomplete in the pre-B, B-cell, and T-cell cell lines. No statistically significant reduction in the *NF-κB p65* transcription factor was noted in the CML cell line. The *NF*-*κB p65* transcription factor is significantly downregulated by SMCAF in a dose-dependent manner in normal B-cells, pre-B ALL, B-cell ALL, and T-cell leukemia cell lines. The CML cells line showed a reduction in *NF-κB p65* that was not statistically significant. Error bars represent the mean (SEM) standard error for three replicates. The dotted line represents the normalized value of 100 for a NALM6 control sample from each blot. Asterisks denote statistical significance relative to its own control (* *p <* 0.05; ** *p <* 0.01, *** *p <* 0.001). For each cell line, Western blot images include the control bands in a separate box from the treated bands; these images are from the same blot and exposure but reflect the cropping out of additional lines that were not used in this evaluation.

**Figure 6 ijms-25-10361-f006:**
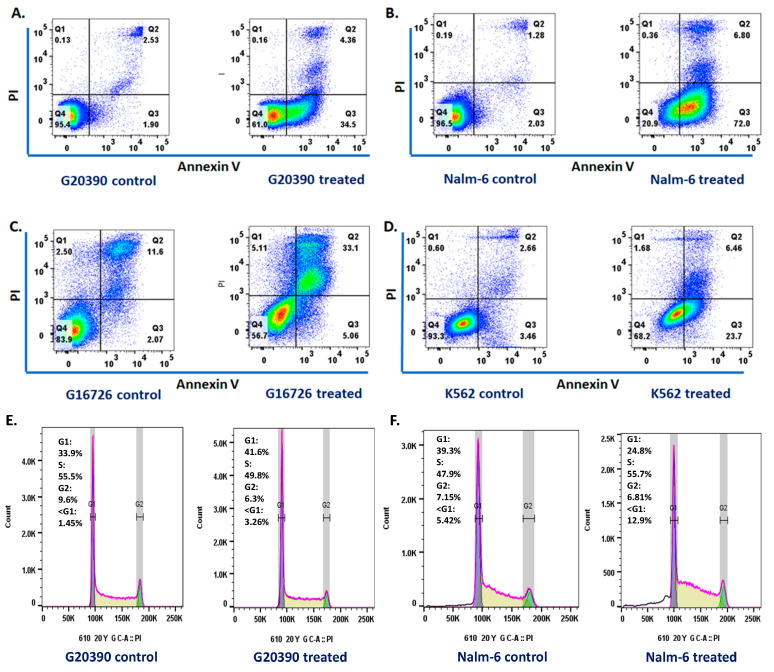
To demonstrate the effects of SMCAF on cell cycle and apoptosis, a single study was performed using GM16726, an acute lymphoblastic leukemia cell line with lymphomatous features: (**A**–**D**) annexin/PI analysis; and (**E**,**F**) cell cycle analysis of leukemic cells after 72 h of treatment with SMCAF. GM16726 is an acute lymphoblastic leukemia cell line with lymphomatous features isolated from a 27-year-old Black patient in the leukemic phase of B-cell lymphoma. The information provided is based on a single investigation.

## Data Availability

The data that support the findings of this study are available from the corresponding author upon request.

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
