# Peer review of "Mycovirus-Containing *Aspergillus flavus* Alters Transcription Factors in Normal and Acute Lymphoblastic Leukemia Cells"

_ijms, 2024, doi:10.3390/ijms251910361_

Round 1

Reviewer 1 Report

Comments and Suggestions for Authors

This manuscript describes the effects of the supernatant of a mycovirus-containing Aspergillus flavus (SMCAF) on the levels of several transcription factors, apoptosis and cell cycle phase distribution in pre-B and B-cell acute lymphoblastic leukemia (ALL) cell lines compared to other leukemia and control cell lines. The presented data are interesting, however there are several concerns about the experimental approach, data presentation and interpretation in the manuscript.

1. Line 94: “This organism was isolated from the home of a patient with ALL…”

- Additional details need to be provided, e.g., Was the MCAF used in this study isolated by the authors? When was it isolated?

2. Line 163: “The levels of transcription factors PAX-5, Ikaros 55 kD and 75 kD and NF-κB p65 were measured, with and without exposure to SMCAF, using immunoblotting technique”

- The data shown in Figs. 2-5 do not look like Western blotting results. What is the “Fluorescence Intensity” on the Y-axes? In any event, original Westrn blot images accompanied by a densitometric analysis should be presented in the manuscript. Did the authors use any protein loading control?

3. Line 268: What is the meaning of this subtitle: “3.2 PAX-5 transcription factor”? There is no description of PAX-5 analysis in this section.

4. The data shown in Fig. 6 should be considered as preliminary results. The data are not sufficient for a statistical evaluation, and thus cannot be properly interpreted. More experiments should be done and the “normal” control cell line should be presented. What is the G16726 cell line?

5. Line 20: “The genetic alterations caused by MCAF are novel findings”.

- No genetics data are presented in the manuscript.

Comments on the Quality of English Language

Minor editing and extensive proofreading are needed

Author Response

We sincerely appreciate the reviewers' comments, which we feel significantly improve the quality of our manuscript. All reviewer suggestions have been accepted, and the manuscript was changed accordingly.

Reviewer 1

This manuscript describes the effects of the supernatant of a mycovirus-containing Aspergillus flavus (SMCAF) on the levels of several transcription factors, apoptosis and cell cycle phase distribution in pre-B and B-cell acute lymphoblastic leukemia (ALL) cell lines compared to other leukemia and control cell lines. The presented data are interesting, however there are several concerns about the experimental approach, data presentation and interpretation in the manuscript.

  1. Line 94: “This organism was isolated from the home of a patient with ALL…”

- Additional details need to be provided, e.g., Was the MCAF used in this study isolated by the authors? When was it isolated? The MCAF was isolated by one of the authors form the home of a patient with ALL several years ago and has been cultured approximately every four weeks. To ensure the persistence of mycovirus within the Aspergillus flavus, the cultures are tested for aflatoxin production and by electron microscopy on a periodic basis.

  1. Line 163: “The levels of transcription factors PAX-5, Ikaros 55 kD and 75 kD and NF-κB p65 were measured, with and without exposure to SMCAF, using the immunoblotting technique”

- The data shown in Figs. 2-5 do not look like Western blotting results. What is the “Fluorescence Intensity” on the Y-axes? In any event, original Westrn blot images accompanied by a densitometric analysis should be presented in the manuscript. Did the authors use any protein loading control? The actual Western blot images are added to each figure.

  1. Line 268: What is the meaning of this subtitle: “3.2 PAX-5 transcription factor”? There is no description of PAX-5 analysis in this section.

The description of PAX-5 is given in the introduction and discussion. Pax5 transcription factor, also known as B-cell specific activator protein (BSAP), is a master regulator of normal development, differentiation, maturation and maintenance of B-cells and is frequently mutated in Deletion or inactivating mutations of Pax5 results in chromosomal rearrangements, translocations and cell arrest. Mutation of PAX-5 is one of the consistent genetic alterations found in B-cell acute lymphoblastic leukemia (B-ALL).

  1. The data shown in Fig. 6 should be considered as preliminary results. The data are not sufficient for a statistical evaluation, and thus cannot be properly interpreted. More experiments should be done and the “normal” control cell line should be presented. What is the G16726 cell line?

The data presented in figure 6 is an example of the findings.in a graphic manner.A sentence is added to clarify this matter. It should be noted that several tests done have produced similar results.

GM16726 is a commercially available acute lymphoblastic leukemia cell line with lymphomatous features isolated from a 27 years old Black patient in leukemic phase of B-cell lymphoma. Th information is added to the text.

  1. Line 20: “The genetic alterations caused by MCAF are novel findings”. Using the dictionary definition of transcription factors i.e. “Transcription factor, molecule that controls the activity of a gene by determining whether the gene’s DNA is transcribed into RNA. Transcription factors control when, where, and how efficiently RNA”, MCAF alters the genetics of the cells.

- No genetics data are presented in the manuscript.

Comments on the Quality of English Language

Minor editing and extensive proofreading are needed

Reviewer 2 Report

Comments and Suggestions for Authors

Infection by fungi such as Aspergillus flavus can lead to serious complications in patients suffering from acute leukemia. The toxicity of the therapies and the associated immunosuppression of the patient can significantly reduce life expectancy, especially in transplanted patients. The authors investigated the influence of mycoviruses released by infected fungal cells on the expression of transcription factors in different cell lines.

The hypothesis is interesting and could become interesting for the scientific community after revision of the manuscript.

1. My critique focuses on the exact quantitative and qualitative description of SMCAF. What is the number of virus particles? Can an MOI be determined? What exactly leads to the deregulation of the transcription factors? Can the authors rule out the possibility that other ingredients influence the expression of the transcription factors?

2. The text should be significantly shortened.

3. The axis labels of the figures should be clarified.

4. The scale is missing on the EM images.

Author Response

Reviewer 2

Submission Date

24 June 2024

Date of this review

17 Jul 2024 22:01:43

Infection by fungi such as Aspergillus flavus can lead to serious complications in patients suffering from acute leukemia. The toxicity of the therapies and the associated immunosuppression of the patient can significantly reduce life expectancy, especially in transplanted patients. The authors investigated the influence of mycoviruses released by infected fungal cells on the expression of transcription factors in different cell lines.

The hypothesis is interesting and could become interesting for the scientific community after revision of the manuscript.

  1. My critique focuses on the exact quantitative and qualitative description of SMCAF. What is the number of virus particles? Can an MOI be determined? What exactly leads to the deregulation of the transcription factors? Can the authors rule out the possibility that other ingredients influence the expression of the transcription factors?

The comment of the reviewer is appreciated and is the subject of future studies. The purpose of the present study was to examine the entire products of mycovirus-containing Aspergillus flavus (SMCAF), since individuals are exposed to the combination. Separation and testing of the mycovirus from its host is the subject of our present investigations.

Regarding ingredients influencing the results, for all controls, exact culture media which was used for the culture of mycovirus-containing Aspergillus flavus was utilized. Culture media did not affect the transcription factors. This information is added more clearly to the manuscript.

  1. The text should be significantly shortened.

Please see the edited text.

  1. The axis labels of the figures should be clarified.

Please see the new graphs

  1. The scale is missing on the EM images.

The scale is the  bottom of each graph.

Round 2

Reviewer 1 Report

Comments and Suggestions for Authors

Comments on the Quality of English Language

Some of the expressions in the text are unclear due to incorrectly constructed phrases. 

Reviewer 2 Report

Comments and Suggestions for Authors

The authors have adapted the manuscript according to the reviewers' suggestions. The manuscript can be published in its current form.

Round 3

Reviewer 1 Report

Comments and Suggestions for Authors

The authors have further improved the manuscript. There are still several minor issues, as follows.

1) Line 14: “…Aspergillus flavus (MCAF) isolated from a patient’s home with ALL”…

- The original expression “…isolated from the home of a patient with ALL..” was clearer.

2) Line 21: “The genetic alterations caused by MCAF are novel findings”.

- No genetics data are presented in the manuscript.

Response: Using the dictionary definition of transcription factors i.e. “Transcription factor, molecule that controls the activity of a gene by determining whether the gene’s DNA is transcribed into RNA. Transcription factors control when, where, and how efficiently RNA”, MCAF alters the genetics of the cells.

Comment: The authors should understand that such textbook knowledge cannot replace the need for a basic assessment of transcription factor target genes to prove “the genetic alterations” in this specific study. Since no direct evidence is presented, they may only SUGGEST that their findings support possible “genetic alterations” by MCAF.  

3) The titles of the legends to Figures 2-5 are exactly the same, which is confusing. I recommend revising these using the titles that appear above the respective Figures. Also, the Figure 6 legend does not have a title.

Comments on the Quality of English Language

Minor editing and thorough proofreading are required.

Author Response

Dear Reviewer,

Thank you for your diligent attention and effort to this work.

“The genetic alterations caused by MCAF are novel findings” is changed to "The noted alterations in the transcription factors caused by MCAF"

Aspergillus flavus (MCAF) isolated from a patient’s home with ALL" is changed to ”Aspergillus flavus (MCAF) isolated from the home of a patient with ALL"

The corrected article with the two changes, as above is attached. 
